# Global Research Trend and Bibliometric Analysis of Current Studies on End-of-Life Care

**DOI:** 10.3390/ijerph191811176

**Published:** 2022-09-06

**Authors:** Genevieve Ataa Fordjour, Amy Yin Man Chow

**Affiliations:** 1Jockey Club End-of-Life Community Care Project, The University of Hong Kong, Hong Kong; 2Department of Social Work and Social Administration, The University of Hong Kong, Hong Kong

**Keywords:** End-of-Life Care, palliative care, advance care, terminal care, bibliometric analysis, research trend

## Abstract

The growing emphasis on evidence-based practice has led to a need for more research on healthcare disciplines, and for the synthesis and translation of that research into practice. This study explored the global research trend in regard to End-of-Life Care (EoLC), and assessed the impact and influence, on the scientific community, of relevant EoLC publications EoLC. Over 350,000 related publications on EoLC were retrieved from three databases (PubMed, Scopus, and Web of Science). Our analysis of the global research trend revealed an exponential rise in the number of related publications on EoLC since the year 1837. This study assessed the bibliometric information of 547 current journal publications on EoLC, sorted by relevance, from the three databases. The USA (47.3%) and the UK (16.1%) were the most productive countries, in terms of the number of relevant publications. The bibliometric analysis also revealed which EoLC research was most impactful and influential, from different parameters including documents, authors, sources, and organisations. The keyword analysis further suggested the growing importance of advance care planning and decision-making in regard to EoLC, as well as an episodic upsurge of EoLC publications related to the COVID-19 pandemic. There were few collaborations among the prolific research on EoLC. This study recommends increased research collaboration across the globe, for wider wisdom-sharing on EoLC issues.

## 1. Introduction

End-of-Life Care (EoLC) is culturally and contextually relevant, and has received substantial interest among researchers and industry practitioners. Many countries have promoted the implementation of EoLC, to varying degrees, according to their economic status, resources, and energy [1,2,3,4,5]. Literature review studies have been accepted as an effectual means to map the existing intellectual territory [6,7]. The review works on EoLC research have been done from various perspectives. For instance, in recent years, Yoong, S.Q. et al. [6] conducted a scoping review, to explore death doulas as supportive companions in EoLC. Syvyk, S. et al. [7] also conducted a systematic review and meta-analysis of studies on colorectal cancer disparities across the continuum of cancer care. Most of these reviews were for synthesising practice directions in the available literature. The Lien Foundation developed an index to rank the quality of death across countries, from the experts’ views [8,9]. The index was re-named and refined as the ‘Quality of Death and Dying Index’ [10], and was compared across geographical regions. These cross-country comparisons were mainly from the service-provision perspective, and there have been few comparisons on research and publications across countries and various parameters.

The cumulative increase in the publication of studies on EoLC necessitated an assessment of the impact and influence of these publications on the scientific community. An assessment of the current articles, using a mapping approach, would provide a considerable amount of knowledge on the trend and status of the research domain. One of the quickest ways to perform an assessment review of a considerable number of publications is through bibliometric analysis [11]. Bibliometric analysis helps to provide an understanding of current research trends in any field of study, and of the various network relationships, such as authors’ citation and collaboration relationships [12]. However, relatively few efforts have been made to map related literature on EoLC by assessing the impact or influence of publications on EoLC. 

A recent publication by Abu-Odah et al. [13] conducted a bibliometric analysis and mapping review of global palliative care research from 2002 to 2020. This study retrieved 19,199 English articles published from the Web of Science and Scopus databases. Another recent study, [14], conducted a bibliometric analysis of global research on the palliative care landscape in the COVID-19 era. This study was limited to publications from 1 January 2020 to 25 April 2022, and retrieved 673 Journal articles, published in various languages, from the Scopus database [14]. Due to the complexity of combining data from multiple databases for bibliometric analysis, it is very common for studies of this nature to employ one or two databases for the data search. The most commonly used database is Scopus, which covers a wider range, and also includes 100% data on Medline [13]. Web of Science and PubMed are also compatible with bibliometric analysis. 

This study sorted for relevant publications on EoLC from the three databases—PubMed, Scopus, and Web of Science. A global research trend on the publications related to EoLC was explored. This study employed bibliometric analysis to assess current relevant publications on EoLC, in order to gain insight into the most influential and impactful documents, authors, countries, sources, and organisations involved in the said field of research. A time period restriction, to publications from 2020 onwards, was set, in order for the bibliometric analysis to explore the current research direction of EoLC, when taking into consideration the influence of the COVID-19 pandemic. The final selection of studies was based on the relevance-ranking of studies on EoLC, using impact metrics from the three databases. 

Thus, the objectives of this study were: (1) to determine the growth research trend of the journal publications on EoLC; (2) to quantitatively assess the contribution of the most relevant literature on EoLC, from different bibliometric parameters; and (3) to identify major themes in EoLC research, using keywords co-occurrence analysis. The findings from this study will be relevant to multitudinous groups of EoLC research stakeholders, such as field practitioners, researchers, and journal editors; in particular, they will inform researchers of the prolific EoLC research across the globe, in the interests of potential international research collaboration on the subject area. Information on the highest impact journals also provides researchers with useful information for future submission on EoLC research. 

## 2. Methodology

### 2.1. Data Source and Selection

This study was conducted by retrieving related publications on EoLC from the PubMed, Web of Science, and Scopus databases, to ensure a reliable coverage of relevant studies. The data retrieval from the 3 databases was conducted on 19 August 2022. 

The search query, using ‘All fields/TITLE-ABS-KEY’, included the following terms: the main keywords ‘End-of-Life Care’ and alternative keywords (‘end of life care’, ‘palliative care’, ‘palliative medicine’, ‘hospice care’, ‘terminal care’, ‘life-limiting’, ‘life threatening’, ‘advance care planning’, ‘life support care’, and ‘incurable disease’) combined with the Boolean operator ‘OR’. No restriction was initially set. This returned a total number of 415,911 publications from PubMed, followed by 400,711 publications from Scopus, with the smallest number, of 355,689 publications, being from Web of Science. The growth trend analysis of publications on EoLC was explored, using these initial findings from the 3 individual databases. Further screening was done for the bibliometric analysis. To ensure the homogeneity and robustness of the analysis, the document type was set to Journal articles only. In order to explore current publications only, a time period was restricted to the year 2020 and onwards. This reduced the number of documents in each dataset (PubMed: 84,755; Scopus: 53,954; and Web of Science: 72,044). Each of these databases provided different impact metrics. To explore the most relevant studies, the first 200 articles, sorted by relevance in the 3 databases, were downloaded and merged in Excel. The final number of documents, after removing duplicates, was 547 articles. These included 173 articles from the year 2020, 218 articles from the year 2021, and 156 articles from the year 2022. Bibliographical information about the 547 articles was exported from the Scopus database. Figure 1 presents the search process for the selected studies.

### 2.2. Research Trend Analysis

This study sought to explore the research trend of studies on EoLC. Using descriptive analysis, the yearly number of journal publications on EoLC, as well as the cumulative number of publications, were assessed to provide a clear estimate of the growth trend in research on EoLC. The current number of publications was also used to determine the most productive countries, institutions, authors, and journal sources in the field of EoLC research. 

### 2.3. Bibliometric Analysis

This study sought to determine the top 20 most impactful authors, documents, and journal sources on EoLC that had been published to date, as well as to determine the various network relations, using bibliometric analysis. Previous studies had presented the top 15 [13] and top 10 [14] parameters from the bibliometric analysis of related studies. The current research themes on EoLC were also explored. The bibliometric analysis was conducted using VOSviewer software (Leiden University, Leiden, The Netherlands). Using the VOSviewer software, different bibliometric maps were generated from the 547 included studies, and analysed, based on citation analysis and co-occurrence analysis of the keywords. The citation analysis was used to reveal the relational networks that existed among the 547 included studies, and to assess the impact of each study in the scientific community, based on the number of times other authors had mentioned it in their work. The network from the citation revealed who had cited whom, and was used to reveal the citation relationships between authors and documents. The co-occurrence-of-keywords analysis was used to reveal the current research themes, based on the most active keywords explored in the studies on End-of-Life Care. 

## 3. Results and Discussions

### 3.1. Results from Research Trend Analysis

#### 3.1.1. Growth Trend of Journal Publications on End-of-Life Care

The number of publications on a yearly basis was assessed from three databases (PubMed, Scopus, and Web of Science), to provide a clear estimate of the research growth trend in the subject area. Using descriptive analysis, the yearly number of publications on End-of-Life Care, as well as the cumulative publications on a year-to-year basis for each database, were obtained, and are shown in Figure 2.

The plotted line graph of the cumulative publications in Figure 2 depicts an exponential growth trend of research publications on EoLC. Different results were obtained from the three databases. The results from Scopus revealed that publications on EoLC date back to the year 1837, whereas PubMed and Web of Science indicated the later years of 1850 and 1970, respectively. EoLC-related studies were published yearly from 1837 to 1963 and, though there were few years within that period in which less than 100 such studies were published, it was from 1990 onwards that researchers showed great interest in EoLC-related studies, with over 1000 publications yearly. In the year 2021, the number of published journal articles on EoLC was the highest to date (Web of Science: 38,708; PubMed: 37,552; Scopus: 29,645). This year, 2022, as of 19 August, about 20,000 journal articles have already been published. The findings extended to the year 2023, with 13 articles identified in Scopus database, 7 from Web of Science, and 4 from PubMed. The findings included journal publications, conferences, books, and book chapters. The results indicate that researchers across the globe still have great interest in studies on EoLC. This is evident from the exponential rise in the number of cumulative journal publications since the year 1990.

#### 3.1.2. Most Productive Countries for Current Research on EoLC

The 547 most relevant journal publications on EoLC were published by 63 countries across the globe. The study identified the top 20 most productive countries that have currently published more than five relevant journal articles on EoLC, from the year 2020. Table 1 presents the list of the most productive countries, their number of article publications, the percentage distribution of the publications, and the World Bank rating of the income levels of each of the top countries. 

The highest number of current relevant Journal publications on EoLC (200 articles, 36.6% of the total documents) was from the United States of America. This was followed by 102 articles (18.6%) from the United Kingdom. Subsequently, Australia and Canada published 54 articles (9.9%) and 52 articles (9.5%), respectively. 

It is evident from Table 1 that all the top 12 countries working on EoLC have high income levels [15]. This is indicative of the fact that economically developed countries have identified the benefits of EoLC research, and are exploring its various dimensions. 

#### 3.1.3. Top Organizations with Highest Current Publications on EoLC 

The study revealed 160 organizations affiliated with the 547 recent publications on EoLC, out of which, 20 top organizations with more than 9 journal publications on EoLC, from the year 2020, were identified. Table 2 presents the list of the top organizations, with their country of location and the number of current relevant publications on EoLC. 

It was found that the University of Toronto in Canada had published the highest number (34) of journal articles, out of the 547 included studies on EoLC from 2020 to date. This was followed by the University of Pennsylvania and the University of Washington, both in the United States of America, with 30 and 19, respectively, of the most relevant journal articles on EoLC. These results also indicate that three countries—the United States of America, Canada, and the United Kingdom—play dominant roles in the research on EoLC globally.

### 3.2. Results from Bibliometric Analysis of Publications from 2020

#### 3.2.1. Citation Analysis of the Documents 

The citation analysis of the documents provided information on the quality of the published document. A publication with a higher citation metric usually indicated that the quality of the document was very good, as it had been cited by many researchers. The bibliometric analysis revealed that, out of the 547 documents under study, only 8 documents had citation connections, and formed three main clusters.

This study further sought to identify the highly cited documents from the 547 current journal publications on EoLC. Table 3 presents the top 20 documents that have been cited more than 11 times.

The top 20 most-cited document was by Lovell N., Maddocks M., Etkind S.N. et al., 2020 [16], which postulated that palliative care is an essential component of the COVID-19 response, and aided care teams to rapidly adapt with the new ways of working. The second-most-cited document was authored by Van den Block L., Honinx E., Pivodic L. et al., 2020 [17], which investigated the effect of Palliative Care for Older People (PACE), for evidence on how to improve palliative care in nursing homes. 

#### 3.2.2. Authors and Co-Authors Analysis

The analysis of the authors and co-authors was done to identify the major research scholars in the field of EoLC. The bibliometric analysis revealed that the 547 documents were written by 2920 authors and co-authors. Table 4 presents the list of 20 top authors and co-authors with more than three relevant article publications on EoLC. The results indicate that these are the most impactful and most influential researchers in the current research on EoLC. 

#### 3.2.3. Source Distribution and Citation Relationship

The analysis of the distribution of EoLC publications among the various journal sources, and their citation relationship, provided an indication of the top journals where the researchers preferred to publish their research on EoLC. The bibliometric analysis revealed 212 sources from the 547 documents under study. Table 5 enlists the 20 top journals, number of citations, total link strength, and their impact factor.

The result indicates that the researchers’ preferred choice for publishing relevant studies on EoLC was the American Journal of Hospice and Palliative Medicine (JIF-2.090). From this source, 42 of the most relevant articles on EoLC have been published, from 2020 to date. These documents have so far accumulated the highest number of citations (78) and the highest total link strength (2). In second position was the Journal of Pain and Symptom Management (JIF-5.576), where 40 of the most relevant articles on EoLC were identified, with the highest accumulated citation counts (339) and total link strength (8). In third, fourth, and fifth positions were Journal of Palliative Medicine (JIF-2.947), Palliative Medicine (JIF-5.713), and BMC Palliative Care (JIF-3.113). The impact factors of the six top journals were above 2.000; this indicates that high quality articles were published in these journals. 

#### 3.2.4. Identifying the Most Active Keywords of EoLC Research 

Co-occurrence analysis of the authors’ keywords was employed, to identify the most common keywords that were used by the EoLC researchers. The bibliometric analysis revealed that there were 1215 keywords used in the 547 documents. Figure 3 presents the network visualisation map of the keywords’ co-occurrence for the 20 most common keywords with a minimum of 10 occurrences.

The science mapping of the 20 most active keywords, with at least 10 occurrences, generated three clusters, with each representing the domain that was most relevant: these included palliative care, end of life care, and advance care planning. The statistical details of the most active keywords have been presented in Table 6.

The results indicate that the most repeated keywords were ‘Palliative care’, followed by ‘End-of-Life Care’. The total link strength, which defines the strength of inter-relatedness between keywords, also revealed ‘Palliative care’ and ‘End-of-Life Care’ as the most active keywords in this research field. 

## 4. Discussion

This paper discusses the global research trend of, and bibliometric information on, the current most relevant articles on EoLC. Three databases—PubMed, Scopus, and Web of Science—were used to retrieve related articles on EoLC. It was observed that there has been an exponential rise in publications on EoLC since it first began in the year 1837, with a cumulative number of over 350,000 publications. The exponential rise in related publications on EoLC was also reported in previous studies [13,36]. The growing worldwide demand for care at the end of life, due to the ageing population, saturation in hospitals, and high prevalence of chronic illnesses [37], may account for the increasing research interest in this field.

A total of 547 of the most relevant articles on EoLC, from the year 2020 onwards, were sorted from the three databases. Bibliometric analysis, using VOSviewer Software, was performed on the information extracted, from the Scopus database, about the 547 articles. The results from the bibliometric analysis revealed that most of the countries involved in EoLC research were economically developed. The growing provision of palliative care in the health systems of several high-income countries may account for the great research interest in this field [38]. The United States of America was the country that had published the highest number (200) of the most relevant current articles on EoLC. This was followed by the United Kingdom. The results revealed that the University of Toronto, Canada, was the most productive affiliated organisation that had published the highest number of the most relevant articles on EoLC. The findings indicate that the United States of America, the United Kingdom, Canada and Australia, are the most productive countries of EoLC-related publications. Despite its having had no language limit, the study’s findings are congruent with a previous study by Adu-Odoh et al. [13], which was limited to English language only, a factor which the authors had reported as having been a possible influence on their study findings. 

The document by Lovell N., Maddocks M., Etkind S.N. et. al. 2020, was the most cited publication. Most of the current relevant articles on EoLC were published by the American Journal of Hospice and Palliative Medicine and the Journal of Pain and Symptom Management. The collaboration connection among the prolific authors and co-authors in this research field was found to be weak, as also reported in similar studies [13,14]. International research collaboration is essential for helping to strengthen the quality of any research work [11].

It was found that the most active keywords in this research domain were ‘Palliative care’, ‘End-of-Life Care’, and ‘Advance care planning’. Though it is not a surprising finding, we can see that the term ‘End-of-Life Care’ exists in different derivatives, such as ‘End-of-Life’, ‘End of Life’, and ‘End of Life Care’. Similarly, the different forms of care for patients with life-threatening illnesses, such as ‘Palliative care’, ‘Hospice’, ‘Hospice care’, and ‘Terminal care’ were referred to, but ‘Palliative care’ was a more frequently used keyword in all the academic publications. Notably, Advance care planning, Communication, and Decision-making are increasingly popular themes in regard to EoLC [14]. Similarly, the term ‘Dementia’ is also popular in EoLC research. The keywords ‘Quality of life’, are a term which is usually referred to as the outcome of End-of-Life Care, and which also appears to be an important concept. These are the areas expecting to receive more research attention. An incidental finding is the popularity of COVID-19 in current EoLC publications. The high death toll of this infectious disease probably alerts patients and families to the need for EoLC. 

There were limitations to this analysis, as only the first 200 publications, sorted by relevance from each of the three databases of PubMed, Scopus, and Web of Science, were included in the bibliometric analysis. Notwithstanding, the high co-occurrences of the keywords ‘End-of-Life Care’ with ‘Palliative care’ and ‘Hospice care’ confirmed the inclusion of most of the relevant publications. The sample size of 547 journal articles was also a limitation, though it was not easy to appraise the large number of papers, which were combined from multiple databases. Publications in all languages were included in the selection process, to capture the global essence of studies around the world. The time period restriction, to publications from 2020 onwards, may also have limited the search output, but to a more manageable level. 

## 5. Conclusions

The results from this study give an indication of the impact and influence of the most relevant recent publications on EoLC in the scientific community. EoLC-related topics have attracted substantial research interest across the globe. The bibliometric analysis revealed the various relevant topics linked with EoLC. The global response to the COVID-19 pandemic has sparked research interest in COVID-19-related EoLC studies. Further studies on COVID-19-related EoLC studies should be extended, to explore the changes in care plans, and directions for End-of-Life Care improvement. This study also affirms the need for international cross-collaboration among prolific researchers in this field, to expand knowledge on EoLC issues globally. 

## Figures and Tables

**Figure 1 ijerph-19-11176-f001:**
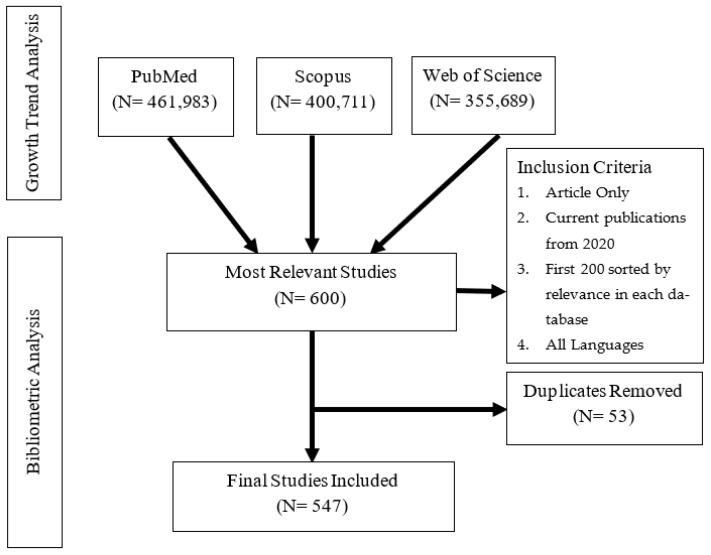
Search process for the selected studies on EoLC.

**Figure 2 ijerph-19-11176-f002:**
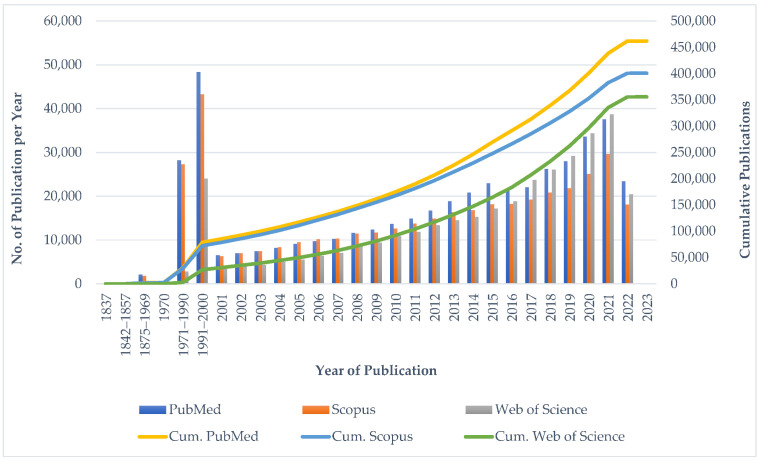
Growth Trend of Journal Publications on EoLC from three databases.

**Figure 3 ijerph-19-11176-f003:**
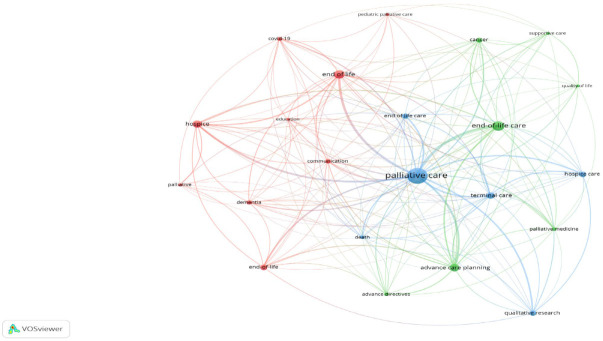
Network visualization map of the keywords’ co-occurrence.

**Table 1 ijerph-19-11176-t001:** Top 20 Countries with the most relevant publications on EoLC.

Rank	Country/Territory	Continent	No. of Articles Published (N = 547)	Distribution of Publications (%)	World Bank Rating (Income Level) [15]
1	United States of America	Americas	200	36.6	High Income
2	United Kingdom	Europe	102	18.6	High Income
3	Australia	Oceania	54	9.9	High Income
4	Canada	Americas	52	9.5	High Income
5	Netherlands	Europe	21	3.8	High Income
6	Germany	Europe	19	3.5	High Income
7	Taiwan, China	Asia	18	3.3	High Income
8	Italy	Europe	16	2.9	High Income
9	Belgium	Europe	14	2.6	High Income
10	Switzerland	Europe	14	2.6	High Income
11	Spain	Europe	13	2.4	High Income
12	Japan	Asia	12	2.2	High Income
13	China	Asia	11	2.0	Upper Middle Income
14	New Zealand	Oceania	11	2.0	High Income
15	Hong Kong SAR, China	Asia	10	1.8	High Income
16	South Korea	Asia	10	1.8	High Income
17	Brazil	Americas	9	1.6	Upper Middle Income
18	Ireland	Europe	8	1.5	High Income
19	Portugal	Europe	6	1.1	High Income
20	Sweden	Europe	6	1.1	High Income

**Table 2 ijerph-19-11176-t002:** Top Organizations with the highest relevant publications on EoLC.

Rank	Name of Organization	Country	No. of Published Articles (N = 547)
1	University of Toronto	Canada	34
2	University of Pennsylvania	United States of America	30
3	University of Washington	United States of America	19
4	Harvard Medical School	United States of America	17
5	National Taiwan University	Taiwan, China	17
6	VA Medical Center	United States of America	16
7	Dana-Farber Cancer Institute	United States of America	10
8	McMaster University	Canada	14
9	University of Alberta	Canada	14
10	Johns Hopkins University	United States of America	13
11	King’s College London	United Kingdom	13
12	Stanford University	United States of America	13
13	Duke University	United States of America	12
14	Indiana University-Purdue University Indianapolis	United States of America	11
15	University of California, San Francisco	United States of America	11
16	University of Colorado	United States of America	11
17	University of Melbourne	Australia	11
18	Vrije Universiteit Brussel	Belgium	11
19	University College London	United Kingdom	9
20	The University of British Columbia	Canada	9

**Table 3 ijerph-19-11176-t003:** Top 20 articles with the most citations.

Rank	Article Title	Authors	Year	No. of Citations	Ref.
1	Characteristics, Symptom Management, and Outcomes of 101 Patients With COVID-19 Referred for Hospital Palliative Care	Lovell N., Maddocks M., Etkind S.N. et al.	2020	106	[16]
2	Evaluation of a Palliative Care Program for Nursing Homes in 7 Countries	Van den Block L., Honinx E., Pivodic L. et al.	2020	29	[17]
3	Actualizing Better Health and Health Care for Older Adults	Fulmer T., Reuben D.B., Auerbach J. et al.	2021	23	[18]
4	Systems Barriers to Assessment and Treatment of COVID-19 Positive Patients at the End of Life	Pahuja M., Wojcikewych D.	2021	20	[19]
5	Health Care Utilization and End-of-Life Care Outcomes for Patients with Decompensated Cirrhosis Based on Transplant Candidacy	Ufere N.N., Halford J.L., Caldwell J. et al.	2020	19	[20]
6	Health and social care professionals’ experiences of providing end of life care during the COVID-19 pandemic: A qualitative study	Hanna J.R., Rapa E., Dalton L.J. et al.	2021	15	[21]
7	Telehealth Acceptability for Children, Family, and Adult Hospice Nurses When Integrating the Pediatric Palliative Inpatient Provider during Sequential Rural Home Hospice Visits.	Weaver M.S., Robinson J.E., Shostrom V.K. et al.	2020	15	[22]
8	Coping With Trauma, Celebrating Life: Reinventing Patient and Staff Support During The COVID-19 Pandemic	Wei E., Segall J., Villanueva Y. et al.	2020	15	[23]
9	Quality of Palliative and End-of-Life Care in Hong Kong: Perspectives of Healthcare Providers	Wong EL-Y., Kiang N., Chung RY-N. et al.	2020	14	[24]
10	End-of-Life quality metrics among Medicare decedents at minority-serving cancer centers: A retrospective study	Wasp G.T., Alam S.S., Brooks G.A. et al.	2020	14	[25]
11	Dying in times of the coronavirus: An online survey among healthcare professionals about End-of-Life Care for patients dying with and without COVID-19 (the CO-LIVE study)	Onwuteaka-Philipsen B.D., Pasman H.R.W., Korfage I.J. et al.	2021	14	[26]
12	Health Care Costs at the End of Life for Patients with Idiopathic Pulmonary Fibrosis. Evaluation of a Pilot Multidisciplinary Collaborative Interstitial Lung Disease Clinic	Kalluri M., Lu-Song J., Younus S. et al.	2020	12	[27]
13	A Qualitative Study of Pulmonary and Palliative Care Clinician Perspectives on Early Palliative Care in Chronic Obstructive Pulmonary Disease	Iyer A.S., Dionne-Odom J.N., Khateeb D.M. et al.	2020	12	[28]
14	The Compassionate Communities Connectors model for End-of-Life Care: a community and health service partnership in Western Australia	Aoun S.M., Abel J., Rumbold B. et al.	2020	12	[29]
15	Navigating the terrain of moral distress: Experiences of pediatric End-of-Life Care and bereavement during COVID-19	Wiener L., Rosenberg A.R., Pennarola B. et al.	2020	12	[30]
16	Death Attitudes, Palliative Care Self-efficacy, and Attitudes Toward Care of the Dying Among Hospice Nurses	Barnett M.D., Reed C.M., Adams C.M.	2021	11	[31]
17	End-of-Life Care in intellectual disability: a retrospective cross-sectional study	Hunt K., Bernal J., Worth R. et al.	2020	11	[32]
18	Association of illness understanding with advance care planning and End-of-Life Care preferences for advanced cancer patients and their family members	Yoo S.H., Lee J., Kang J.H. et al.	2020	11	[33]
19	Experiences and needs of patients with incurable cancer regarding advance care planning: results from a national cross-sectional survey	Stegmann M.E., Geerse O.P., Tange D. et al.	2020	11	[34]
20	Hospice Palliative Care (HPC) and Medical Assistance in Dying (MAiD): Results from a Canada-Wide Survey	Antonacci R., Baxter S., Henderson J.D. et al.	2021	11	[35]

**Table 4 ijerph-19-11176-t004:** Top 20 Authors with more than three current journal publications on EoLC.

Rank	Author	Number of Articles	Number of Citations	Average Citation Per Article	Total Link Strength
1	Jones C.A.	7	16	2.29	6
2	Cohen J.	6	24	4.00	8
3	Grudzen C.R.	6	42	7.00	11
4	Kaasalainen S.	6	9	1.50	11
5	Weaver M.S.	6	44	7.33	8
6	Deliens L.	5	50	10.00	9
7	Gott M.	5	14	2.80	5
8	Sampson E.L.	5	9	1.80	6
9	Baxter S.	4	14	3.50	6
10	Cuthel A.M.	4	27	6.75	9
11	El-Jawahri A.	4	31	7.75	2
12	Ersek M.	4	12	3.00	3
13	Ko M.-C.	4	6	1.50	13
14	Robinson J.	4	9	2.25	5
15	Robinson J.E.	4	35	8.75	7
16	Sussman T.	4	9	2.25	7
17	Van Den Block L.	4	32	8.00	10
18	Van Der Heide A.	4	16	4.00	3
19	Zimmermann C.	4	33	8.25	0
20	Pasman H.R.W.	4	48	12.00	8

**Table 5 ijerph-19-11176-t005:** Top 20 Journals where the most current relevant article publications on EoLC were published.

Rank	Source	No. of Documents (N = 547)	No. of Citations	Total Link Strength	Journal Impact Factor (JIF) (2021) ^a^
1	American Journal of Hospice and Palliative Medicine	42	78	2	2.090
2	Journal of Pain and Symptom Management	40	323	8	5.576
3	Journal of Palliative Medicine	35	115	5	2.947
4	Palliative Medicine	33	85	9	5.713
5	BMC Palliative Care	30	51	8	3.113
6	BMJ Supportive and Palliative Care	17	46	1	4.633
7	Progress in Palliative Care	15	39	2	0.670
8	Journal of Palliative Care	13	35	3	1.980
9	Journal of Hospice and Palliative Nursing	12	10	1	2.131
10	International Journal of Environmental Research and Public Health	11	23	2	4.614
11	International Journal of Palliative Nursing	9	5	0	0.718
12	BMJ Open	8	19	1	3.006
13	British Journal of Nursing	7	1	0	0.710
14	Omega (United States)	6	0	2	2.602
15	Supportive Care in Cancer	6	29	2	3.359
16	Death Studies	5	15	0	4.340
17	Health and Social Care in the Community	5	9	1	2.395
18	Social Science and Medicine	5	7	0	5.379
19	Palliative and Supportive Care	4	14	0	3.733
20	Palliative Care and Social Practice	4	14	1	4.600

^a^ The Journal Impact Factor (JIF) is the journal-level metric calculated from data indexed in Web of Science Core Collection and Google search, current update 2022.

**Table 6 ijerph-19-11176-t006:** 20 Most active keywords with more than 100 occurrences.

Rank	Keyword	Occurrences	Total Link Strength
1	Palliative care	310	457
2	End-of-Life Care	111	161
3	End of Life	87	138
4	Advance care planning	75	125
5	Hospice	66	118
6	Terminal care	52	103
7	End-of-Life	41	64
8	Hospice care	38	71
9	End of life care	35	68
10	Qualitative research	34	62
11	Cancer	28	50
12	Communication	27	66
13	Dementia	26	54
14	Covid-19	21	47
15	Advance directives	18	40
16	Death	17	49
17	Palliative medicine	16	36
18	Palliative	13	25
19	Quality of life	12	22
20	Supportive care	12	26

## Data Availability

The data employed in this study are available, on reasonable request, from the corresponding author.

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
