# Peer review of "Global Research Trend and Bibliometric Analysis of Current Studies on End-of-Life Care"

_ijerph, 2022, doi:10.3390/ijerph191811176_

Round 1

Reviewer 1 Report

The manuscript deals with a relevant and interesting topic and the aim is commendable. However some methodological issues are present in the literature search that may impact on the robustness of the results. Find below some specific comments for improvement:

- please clarify why PubMed has not been used since the topic is related to healthcare field

- please clarify why year 2020 has been chosen as cutoff

- 7.000 papers are quite a lot in my opinion, probably this is due to the usage of general single keywords and not a constructed search string validated through snowballing referencing. Also the fact that you found only 902 documents with citation connections might support this hypothesis. Probably the search string should be refined with a more comprhenesive set of keywords and/or with some inclusion / exclusion criteria

- A prism diagram could clarify the process and improve clarity/readability for the reder/researcher that might benefit from the study

- Results presented in section 3.1.1 are coherent with growth trend of publications in all sector. Which are the implication for EoLC? Why this trend is relevant for the topic? Probably this growth can be mentioned in the introduction since the total number is not referring to the selected papers but to the general 34.000 results from the first screenings.

- Why 150 citation have been chosen as cutoff in table 3?

- Referring to other Bibliometric analysis (even in other field) can strengthen the methodological process

- Discussion section  is missing

- In general the limitations mentioned in the conclusion chapter with specific regard to the keyword selection significantly impact the overall quality and robustness of the result presented; additionally the impact for the scientific field of the results presented (i.e. table 6) are not clear: how a researcher can benefit from the outcome of this short study? This should be detailed explained in a discussion section

Author Response

Dear Reviewer, we are very thankful for your effort and time in offering us constructive comments.

Reviewer 2 Report

Global Research Trend and Bibliometric Analysis of Studies on 2 End-of-Life Care during Covid-19 outbreak

This study aimed to assess the global trend of studies carried out in End-of-Life Care through a bibliometric analysis.

1) Since this study conducted a systematic review, I strongly recommend that the authors should review the PRISMA guidelines for systematic reviews.

2) When reading the title, one expects that end-of-life care studies would be related to COVID 19. However this is not the case, as it only refers to the publication period of the article. Authors should specify this in the title.

3) In the methods section authors should explain in detail that the 34,943 articles were used to determine the increasingly number of publications on EoLC, while the other 7,154 for the Bibliometric Analysis. The information contained in the tables is is confusing.

4) I suggest that authors should change the subtitle 3.1.2. into 3.1.2. Most Productive Countries of Current Research on EoLC from 2020, while 3.1.3 should change into 3.1.3. Top Organizations with highest current publications on EoLC from 2020 in order to get the picture about the 7154 articles.

5) Although figure 2 is described in the text, the title of the figure below the network visualization map of the keywords co-occurrence is not presented.

Author Response

Thank you very much for your help in reviewing the manuscript

Reviewer 3 Report

The subject of the present study is relatively important and new. However, this research contains major concerns mentioned below.

Abstract

  • In lines 18-20; Instead of referring to the general sentence "The results from the bibliometric analysis gave an indication ….", the results in these areas should be referred to in summary.
  • Considering the focus of the study during the COVID-19 outbreak, COVID-19 could be added in keywords.

Introduction: 

The sentence stated in lines 32-33 “Literature review studies have been accepted as 

  •  effectual…” should be referenced.
  • The necessity of limiting the study to the period during the outbreak of Covid-19 should also be stated in the introduction section.

Methods:

  • I agree about the comprehensive coverage of articles in the Scopus database. However, I believe that if you search in databases such as PubMed and Web of Science, the validity of the search and the assurance of access to all relevant articles would be much higher than searching only in the Scopus database. Because some articles are only indexed in the PubMed database and some are only indexed in the Web of Science, they will not be found in the Scopus database search. Therefore, I believe that searching in the Scopus database alone is a major concern of this study and it reduces the quality of the study.
  • The search keywords were not selected in a standard way, which could affect the quality of the retrieved records. For example, the preferred keyword for “end of life care” is “Terminal Care”. This keyword itself also has synonyms that can help in finding related literature such as “End of Life Care “, “End-Of-Life Care”, “Advance Care Planning”, “Hospice Care”, “Life Support Care”, and “Palliative Care”.
  • Why did the authors limit the study to during the COVID-19 outbreak period? Did they have a special justification for doing this?
  • Why did the authors limit the study to the period of Covid-19? Did they have a special justification for doing this?

Results:

  • I suggest that the results section and the discussion are separated from each other and presented in two separate main sections so that the reader can have a better understanding of the results and discussion.
  • What were the criteria for presenting the records of each of the tables in the results section? For example, in Table 1., the first 10 countries with the highest frequency of publication of articles are presented. Why are more records not provided? Because a uniform trend is not considered for the number of records in all tables.
  • I did not see the title of Figure 2 in the results section.

Discussion:

  • It is suggested that the discussion section be presented separately. In the discussion section, the findings of the current research and comparison with other studies should be discussed.
  • It is suggested to present the strengths and limitations of the study at the end of the discussion section.
  • Authors' recommendations based on research findings can also be added in the discussion or conclusion section.

Author Response

Dear Reviewer, we are thankful for your help in improving the manuscript.

Round 2

Reviewer 3 Report

The authors properly addressed satisfactorily all of the concerns. Thus, the manuscript has now could be accepted for publication in the Applied Clinical Informatics journal.